# A Novel Mouse Model of TGFβ2-Induced Ocular Hypertension Using Lentiviral Gene Delivery

**DOI:** 10.3390/ijms23136883

**Published:** 2022-06-21

**Authors:** Shruti V. Patil, Ramesh B. Kasetti, J. Cameron Millar, Gulab S. Zode

**Affiliations:** Department of Pharmacology and Neuroscience, North Texas Eye Research Institute, University of North Texas Health Science Center, Fort Worth, TX 76107, USA; shrutipatil@my.unthsc.edu (S.V.P.); ramesh.kasetti@unthsc.edu (R.B.K.); cameron.millar@unthsc.edu (J.C.M.)

**Keywords:** TGFβ2, lentiviruses, trabecular meshwork, intraocular pressure, rodent mouse models of ocular hypertension

## Abstract

Glaucoma is a multifactorial disease leading to irreversible blindness. Primary open-angle glaucoma (POAG) is the most common form and is associated with the elevation of intraocular pressure (IOP). Reduced aqueous humor (AH) outflow due to trabecular meshwork (TM) dysfunction is responsible for IOP elevation in POAG. Extracellular matrix (ECM) accumulation, actin cytoskeletal reorganization, and stiffening of the TM are associated with increased outflow resistance. Transforming growth factor (TGF) β2, a profibrotic cytokine, is known to play an important role in the development of ocular hypertension (OHT) in POAG. An appropriate mouse model is critical in understanding the underlying molecular mechanism of TGFβ2-induced OHT. To achieve this, TM can be targeted with recombinant viral vectors to express a gene of interest. Lentiviruses (LV) are known for their tropism towards TM with stable transgene expression and low immunogenicity. We, therefore, developed a novel mouse model of IOP elevation using LV gene transfer of active human TGFβ2 in the TM. We developed an LV vector-encoding active hTGFβ2^C226,228S^ under the control of a cytomegalovirus (CMV) promoter. Adult C57BL/6J mice were injected intravitreally with LV expressing null or hTGFβ2^C226,228S^. We observed a significant increase in IOP 3 weeks post-injection compared to control eyes with an average delta change of 3.3 mmHg. IOP stayed elevated up to 7 weeks post-injection, which correlated with a significant drop in the AH outflow facility (40.36%). Increased expression of active TGFβ2 was observed in both AH and anterior segment samples of injected mice. The morphological assessment of the mouse TM region via hematoxylin and eosin (H&E) staining and direct ophthalmoscopy examination revealed no visible signs of inflammation or other ocular abnormalities in the injected eyes. Furthermore, transduction of primary human TM cells with LV_hTGFβ2^C226,228S^ exhibited alterations in actin cytoskeleton structures, including the formation of F-actin stress fibers and crossed-linked actin networks (CLANs), which are signature arrangements of actin cytoskeleton observed in the stiffer fibrotic-like TM. Our study demonstrated a mouse model of sustained IOP elevation via lentiviral gene delivery of active hTGFβ2^C226,228S^ that induces TM dysfunction and outflow resistance.

## 1. Introduction

Primary open-angle glaucoma (POAG) is the most prevalent form contributing to nearly 74% of total glaucoma cases worldwide [1,2,3,4]. Elevated intraocular pressure (IOP) is the major risk factor responsible for inducing optic neuropathies in POAG [5,6]. The constant turnover of aqueous humor (AH) contributes to maintaining normal physiological levels of IOP [7]. Aqueous humor is secreted by the ciliary body into the anterior chamber and gets filtered from trabecular meshwork (TM). The blockage of the AH outflow pathway elevates IOP in POAG [8,9]. In humans, the conventional pathway via lamellated TM and continuous Schlemm’s canal (SC) plays the greatest role in regulating AH outflow [10]. The mouse eye is suggested to have a similar conventional outflow mechanism, making the mouse a reasonable model to induce and mimic the glaucomatous alterations observed in the anterior chamber of humans [11]. Several inducible rodent models of IOP elevation have been developed by means of surgical or pharmacological interventions to obstruct the outflow pathway [12]. These animal models may not emulate the clinical attributes of glaucoma in its entirety [13]. To help substantiate certain pathological features and generate a suitable mouse model, TM can be targeted with recombinant viral vectors overexpressing cargo genes associated with glaucoma pathogenesis [14]. TM is a target for in vivo viral gene delivery system since the majority of AH exits from TM’s sieve-like structure.

Excessive deposition and cross-linking of extracellular matrix (ECM) in TM tissue caused by molecular alterations increase conventional outflow resistance in POAG [15,16,17,18,19]. Fibrotic-like stiffening of the TM tissue is concurrent with the actin cytoskeletal rearrangement into structures called crossed-linked actin networks (CLANs) [20,21,22,23]. ECM remodeling and cytoskeletal changes have been linked to the altered production of fibrotic-cytokine, transforming growth factor (TGF)-β2 in the AH and the TM [24,25,26,27,28]. Mouse models of TGFβ2-induced IOP elevation have provided key molecular mechanisms of disease progression [29,30,31]. Pro-TGFβ2, linked to a latency-associated protein (LAP), is expressed in its latent state, later forming a dimeric complex [32]. The independent TGFβ2 molecule (~25 kDa), after being enzymatically cleaved from its LAP motif, binds to the TGFβ-receptor and initiates downstream signaling. Elevated levels of activated TGFβ2 in the aqueous humor are required to induce glaucomatous damage in the TM, as observed in human aqueous humor of POAG patients [33,34,35,36,37,38,39,40]. In recent years, an experimental mouse model of open-angle glaucoma was successfully created via intraocular injection of adenovirus-5 (Ad5) constructs carrying an active human-TGFβ2 (hTGFβ2) gene [41]. The introduction of two-point mutations in the LAP domain promotes expression of spontaneously active TGFβ2, as described in previous studies [42]. The selective tropism of Ad5 in TM with high transduction efficiency and the ability to rapidly elevate IOP by increasing active hTGFβ2 expression in the anterior segment has been successful.

Viral vectors, including adenovirus (AdV), herpes simplex virus (HSV), adeno-associated virus (AAV), or lentivirus (LV), have emerged as potent tools for transducing ocular tissues (primarily TM) over the past two decades [14,43]. The feasibility of using this approach in mice, without the need to generate expensive and laborious transgenic mouse lines, further expands its application. However, the disadvantages of certain viral gene transfer mechanisms may limit their applicability. This includes different humoral or cellular immunogenic responses and transient transgene expression in ocular tissues transduced, especially with Ad5 or HSV vectors [41,44,45,46,47]. AAV vectors have low transduction efficiency towards the TM [48]. Even though the self-complementary AAVs are efficient in TM transduction, their vector cassette has a low gene load capacity [49]. Human or feline immunodeficiency virus (HIV or FIV)-based lentiviruses, a phylogenetic sub-class of ssRNA retroviruses, have large cargo packaging capacity with an ability to genomically integrate into a wide range of dividing, non-dividing, or terminally differentiated cells [14,50]. Compared to other viral systems, LV constructs can stably transduce the TM cells with long-term transgene expression efficiency, low to transient immunogenicity, and negligible changes to the aqueous outflow system [51,52,53,54].

Considering the advantages of lentiviral vectors over other viral gene delivery systems, we reproduced a TGFβ2-induced ocular hypertension (OHT) mouse model using an HIV-based LV cassette. The TGFβ2 gene with the same two-point mutations in the LAP region, as described in Shepard et al. [41], was used to attain the constitutive expression of active TGFβ2 under the cytomegalovirus (CMV) promoter. The goal of this study was to achieve sustained IOP elevation and low immunogenicity via lentiviral gene transfer of active TGFβ2 transgene having TM region specificity. Our findings also report the cytoskeletal and ECM changes in the TM cells induced by active TGFβ2 expression. 

## 2. Results

### 2.1. Intravitreal Injection of Lentiviral Vector Expressing hTGFβ2^C226/228S^ Leads to Sustained IOP Elevation in Mice

To develop a mouse model of TGFβ2-induced OHT with low immunogenicity, we utilized lentiviral particles to express active TGFβ2 levels in TM (Figure 1). First, we determined whether lentiviral particles have specific tropism to TM in mouse eyes. A single intravitreal injection of lentiviral particles expressing eGFP was performed in C57BL/6J (male) mice (*n* = 3; Figure 2). We observed a robust and sustained eGFP fluorescence in TM (two of three injected mice). Next, we examined whether a single intravitreal injection of LV_CMV_hTGFβ2^C226/228S^ (LV_TGFβ2) leads to sustained IOP elevation. C57BL/6J mice (4 months old) were injected intravitreally with LV_TGFβ2 in one eye while the contralateral eyes were injected with LV_CMV_Null (control; LV_Null). IOPs were monitored weekly. After 3 weeks, daytime-conscious IOP measurement demonstrated a significant and sustained increase in IOPs in LV_TGFβ2 injected eyes compared to the control eyes (Figure 3). Delta change in IOP in LV_TGFβ2-treated eyes over control eyes was 3.3 mmHg and stayed elevated up to 7 weeks post-injection. However, we observed some variability in IOP increase wherein 50% of the mice showed a mean difference of 6.19 mmHg (average IOP of the control eyes was 12.82 mmHg versus LV_TGFβ2 eyes was 19.01 mmHg; *n* = 5) and the remaining mice demonstrated less than 2.5 mmHg (*n* = 5) increase in IOP. Similar results were observed in separate groups of animals when IOPs were recorded using isoflurane anesthesia (*n* = 5; Appendix A). The mean IOP change in TGFβ2-treated eyes over control was ~4 mmHg during daytime and ~5 mmHg during nighttime.

We also verified whether LV_TGFβ2 elevates IOP in BALB/cJ mice as this strain was utilized previously for Ad5-based gene delivery [41,55]. In BALB/cJ female mice, LV_TGFβ2 induced significant IOP elevation (*p* < 0.001; Figure 4). Conscious IOP during daytime exhibited an average increase of 2.8 mmHg, inclusive of all mice (*n* = 10). However, after segregating the mice with an IOP increase of >2.5 mmHg, an IOP delta change of 4.12 mmHg was recorded (average IOP of control eyes was 11.15 mmHg versus TGFβ2 expressing eyes was 15.27 mmHg; *n* = 5). 

We assessed whether LV_TGFβ2-induced IOP elevation contributes to functional loss in retinal ganglion cells (RGCs) (Appendix A). Pattern electroretinogram (PERG) derived amplitudes and latencies were measured 8 weeks after injection and compared between LV_Null and LV_TGFβ2 contralateral eyes (Appendix A; data representing one experiment). No significant decrease in PERG amplitudes or delay in latency periods was observed in LV_TGFβ2 eyes compared to their contralateral controls (total *n* = 24 from three experiments; data not shown). We also examined whether LV_TGFβ2 induced OHT contributes to loss of RGCs via RBPMS staining of the whole retina isolated 9 weeks post-injection (Appendix A). Number of RBPMS-positive RGCs were analyzed in an automated and masked manner using ImageJ software from peripheral, mid-periphery, and central regions of whole retinal mounts. Overall, there was no significant RGC loss revealed in our study cohorts.

### 2.2. Decrease of Aqueous Humor Outflow Facility in LV_TGFβ2 Expressing Eyes

An increase in IOP is associated with a reduced outflow facility in POAG. Therefore, we further measured outflow facility in live C57BL/6J mice injected with LV_TGFβ2 and LV_Null vectors (Figure 5). Outflow facility measurements were performed in ocular hypertensive mice (Figure 3B; *n* = 4) at 7 weeks of injections and in non-ocular hypertensive mice (*n* = 5) at 8 weeks of injections. Overall, the mice with elevated IOPs showed a significant 64.05% reduction (*p* = 0.027) in the AH outflow facility (Figure 5B), while the facility measurement in mice with no IOP elevation had no significant difference. The mean outflow facility in the control eyes was 40.03 nL/min/mmHg, considerably higher compared to the mean of 22.48 nL/min/mmHg observed in LV_TGFβ2 infected eyes. These data indicate that LV_TGFβ2 leads to a reduced outflow facility and IOP elevation.

### 2.3. Gross Morphological Assessment of Ocular Tissues in LV-Injected Mice

Since adenoviral injections of TGFβ2 induced ocular inflammation [41,55,56], we determined whether LV_TGFβ2-induced OHT is associated with ocular inflammation or other abnormalities. H&E staining (Figure 6A) demonstrated open-angle and no infiltration of inflammatory cells in both control and TGFβ2-treated eyes. Slit-lamp images (Figure 6B) revealed no sign of anterior inflammation, corneal opacity, or other alterations to the corneal structure in both control and TGFβ2-treated eyes. Direct ophthalmoscopy assessment showed no evidence of immunogenicity in any lentiviral infected eyes at any timepoint of the 8-week experimental course. Ad5-induced TGFβ2 has been shown to cause lenticular opacity in rodent eyes [41,57]. We observed mild development of lenticular opacity in some LV_TGFβ2-injected eyes of C57BL/6J mice (Figure 6C). However, this opacity appearance in the lens occurred 2 weeks post-injection of viral particles and was small, discrete, or barely visible throughout the study. Moreover, this observation had no correlation with the eyes showing an increase in IOP. None of the BALB/cJ mice developed opacities in the lens or hyperemia in the anterior chamber (data not shown), except for one control injected eye. 

### 2.4. TGFβ2 Expression in AH and Anterior Segment Tissue of LV_TGFβ2 Injected Eyes

We next assessed the active TGFβ2 protein levels in AH of injected mouse eyes via Western blot analysis, and then in the anterior segments via immunostaining. AH samples (~4 μL) collected from control and LV_TGFβ2-treated eyes were subjected to Western blot analysis for TGFβ2 (Figure 7A–D). Densitometric analysis showed increased active TGFβ2 protein levels in AH of LV_TGFβ2-treated eyes. Evaluation of TGFβ2 expression from AH at week 11 after injection revealed a significant correlation (*r* = 0.87; *p* < 0.01) with IOPs measured at week 10 post-injection (Figure 7D). In a separate group of animals (Figure 7E–K), the paraffin-fixed anterior segments of LV_TGFβ2-injected eyes showed increased fluorescent intensity of TGFβ2 immunostaining compared to control eyes. Co-staining of the same tissues with alpha smooth muscle actin (αSMA) presented a concurrent increase in αSMA expression in LV_TGFβ2 eyes versus the null eyes. Although our analysis of TGFβ2 in AH (Figure 7B) and at the iridocorneal angle (Figure 7F) were not statistically significant, separate analysis of ocular hypertensive eyes (Figure 7C,G) demonstrated that active TGFβ2 levels were significantly elevated in AH (by 4.78-fold) and in anterior segments (by 93.96%) of LV_TGFβ2-treated eyes. Likewise, αSMA showed an increase of 93.94% in eyes with increased TGFβ2 expression (Figure 7I,J). Positive correlation was observed between IOP (measured at week 8) and anterior segment tissues co-stained with TGFβ2 (*r* = 0.51; *p* = 0.06; Figure 7H) and αSMA (*r* = 0.68; *p* = 0.01; Figure 7K).

### 2.5. LV_TGFβ2 Stimulates ECM Deposition and Actin Reorganization in Primary TM Cells

TGFβ2 is known to induce ECM deposition and actin cytoskeleton changes in primary human TM cells [26]. We examined whether LV_TGFβ2 stimulates ECM deposition and actin changes in primary human TM cells. Primary human TM cells (*n* = 4 strains) were treated with LV_Null or LV_TGFβ2 (5 MOI), and cells were maintained for 11 days. Conditioned media was subjected to Western blot analysis for TGFβ2. As shown in Figure 8A, LV_TGFβ2-treated TM cells demonstrated elevated active TGFβ2 protein levels in the conditioned media. The level of TGFβ2 expression varied between the transduced human TM strains (two out of four strains; Appendix A). Immunostaining revealed increased TGFβ2, fibronectin (FN), and collagen-I (Col-I) in LV_TGFβ2 treated primary human TM cells (Figure 8B). Immunostaining for actin cytoskeleton using Phalloidin and for αSMA demonstrated that LV_TGFβ2 dramatically increased actin stress fibers and CLANs in primary human TM cells.

## 3. Discussion

Glaucoma is a complex disorder, and the development of investigative models can provide insight into its pathophysiology [1,58]. Inducible rodent models are a popular choice for exploring precise pathological mechanisms associated with increased IOP [12,13]. Prolonged genetic transformation of cells regulating the outflow pathway system with specific pathological transgenes is a potent methodology for these studies [14,43,51,53,55]. Genetically modified models hold potential for evaluating therapeutic approaches specifically targeting the molecular mechanism of interest [11,29,31,56,59,60,61]. Of the several glaucoma-associated molecular markers identified, TGFβ2 has an imperative role in POAG progression [24,27,35,62]. TGFβ2 is abundantly expressed in ocular tissues compared to its other isoforms [33,36,37,38]. TGFβ2 levels are elevated in glaucoma eyes [34,39]. In this study, we induced sustained IOP elevation in mouse eye via lentiviral-induced expression of active TGFβ2 in the anterior segment. The lentiviral delivery system was independent of any detectable inflammatory responses or corneal dystrophy. 

The LV recombinant system was incorporated with the same mutated-LAP-motif TGFβ2^C226/228S^ transgene described in the Ad5 delivery-based TGFβ2-mouse model [41]. Ad5_TGFβ2^C226/228S^ significantly elevates IOP compared to Ad5_TGFβ2^WT^ or Ad5-Empty inoculated eyes. However, the Ad5 vector promotes a certain degree of immunogenicity that manifests as anterior chamber shallowing, corneal edema, iridial hyperemia, and the presence of inflammatory cells in the angle [41,56]. Such adverse responses can hinder the appropriate articulation of experimental data. In addition, the Ad5_TGFβ2^C226/228S^ model exhibited transient transgene expression and IOP elevation that may limit its application for studies involving extended glaucomatous TM damage [41,55]. Contrary to these observations, LV-induced TGFβ2 exhibited several advantages over the Ad5 vector. Even though it took 3 weeks for the IOP to rise to statistically significant levels, it was elevated over a period of 8 to 9 weeks post-LV injections (Figure 3, Figure 4, and Appendix A). This observation was reproducible among the experiments conducted and was based on IOP measurements performed on mice in both conscious and unconscious states. In POAG pathogenesis, the altered functionality of TM disrupts aqueous outflow mechanisms and elevates IOP [10,63]. Our studies demonstrate that LV_TGFβ2 induced OHT is likely due to TM dysfunction, which leads to reduced outflow facility (Figure 5). Most importantly, slit lamp and H&E staining demonstrated little or no inflammation due to viral vectors (Figure 6), making this model more suitable for the testing of various therapeutic strategies targeting the TGFβ2 pathway. 

Although our data demonstrate sustained IOP elevation, we did not observe any functional or structural RGC loss as expected in glaucomatous pathophysiology (Appendix A). Since an average delta change in IOP was observed at 4–5 mmHg, it is probably insufficient to induce glaucomatous neurodegeneration in C57BL/6J and BALB/cJ mice. Consistent with this, McDowell et al. also demonstrated that sustained IOP elevation induced by Ad5-based mutant myocilin (mut-MYOC) expression did not cause RGC loss in C57BL/6J, BALB/cJ, and A/J mouse strains [56]. Except that the A/J strain was susceptible to optic nerve damage. Future studies can delineate the genetic cause for such variation in OHT-associated retinal and optic nerve degeneration among different mice strains. Previous studies using microbeads have shown 20% axonal degeneration following 4–5 weeks of a 30% increase in IOP [64]. However, the IOP increase using microbeads is caused by the direct occlusion of AH outflow via TM. IOP elevation is less likely to be affected by physiological or environmental factors in models that occlude or sclerose tissues of a conventional outflow pathway. In contrast, the increased expression of TGFβ2 or other POAG-associated genes via viral transduction may contribute to TM dysfunction, rather than direct TM blockage. Due to differences in mechanisms, it is unreasonable to compare these models. Specifically, there may be other pathological events occurring in the microbead model that predisposes RGCs leading to their death. Importantly, microbead-induced blockage of outflow may raise IOPs acutely within hours of injection that can create ischemia in RGCs. These insults can also trigger an immune response in RGCs, further predisposing them to cell death. Moreover, physiological and environmental parameters can fluctuate IOP in our model and may slow down RGC degeneration. A recent study focused on determining IOP fluctuations exerted by internal physiological processes and environmental disturbances using a wireless telemetry system [65]. We performed our Tonometry measurement every week at a set time of day. This provides limited information on IOP variations occurring over an unrecorded course of time. Moreover, several studies have indicated that AH outflow is segmental and caused by the presence of high and low flow regions in the TM [66]. The transduction of such segmental TM by LV_TGFβ2 can also contribute to variations in IOP. Future studies will investigate the effect of IOP fluctuations on RGC degeneration in OHT rodent models induced by glaucoma-associated genes.

Our LV vector was designed to specifically transduce the TM region of the mouse eye. However, based on the eGFP expression, we found that LV may marginally target other non-specific regions of the iridocorneal angle, such as the cells from the iris or ciliary body tissues (Figure 2). This agreed with the increased TGFβ2 expression observed in the TM and surrounding tissues following LV_TGFβ2 vector injections in the respective mouse eyes (Figure 7E–H). Similar non-TM-specific sporadic expression of transgenes in the anterior segments via LV or other viral vectors have been observed in other studies [67], including studies involving rats [41,51,60], feline [68,69], and primate models [52,70]. Although certain studies have claimed that the VSV-G pseudotype nature of the LV vector could be more specific to the TM, the LV virus tends to have a certain tropism towards the other outflow tissues, possibly due to their endothelial nature or due to minor variations to the batches of vector constructs [51,68,71]. The ubiquitous nature of the CMV promoter could contribute to diverse tissue expression of the transgene. In other gene-editing applications, the activity of the CMV promoter has been shown to spontaneously diminish over an extended period [72]. Nonetheless, CMV-derived transgene expression is reported to be stable in non-dividing cells such as those found in the outflow system. Several previous studies have shown that CMV was able to mediate prolonged expression in TM of older mice (>5–6 months) [31,56]. Such studies have highlighted the likelihood of strain and age differences in the success of the viral-induced transgene expression that causes OHT. Regardless, the CMV promoter prolonged transgene expression in our study and elevated the secreted form of active TGFβ2 in the aqueous humor (Figure 7A–D). The development of ECM deposition and cross-linking has been linked to TGFβ2-induced TM pathology and IOP elevation [15,16,17,18,73]. TGFβ2 is a profibrotic cytokine that plays a central role in inducing OHT and fibrotic response [24,25,27]. Furthermore, TGFβ2-induced CLAN formation is associated with stiffer TM tissue [20,21,28]. The correlation of CLANs to TM cell stiffness was recently demonstrated by Peng et al. using atomic force microscopy [23]. Concomitant to these previous findings, our LV_TGFβ2-transduced primary cells demonstrated increased deposition of ECM proteins, along with increased CLANs, actin stress fibers, and αSMA (Figure 8). Mouse TM tissues with elevated TGFβ2 expression also showed increased αSMA (Figure 7E–K), which has been previously associated with TM tissue stiffness [74]. These findings suggest a link between increased TGFβ2 expression, altered ECM remodeling, and actin changes in TM to the decreased outflow facility and IOP elevation. 

It is interesting to note that we observed only 50% of LV_TGFβ2-treated mice with significant IOP elevation. Analysis of TGFβ2 protein levels in AH in OHT and non-OHT mice further suggested that IOP elevation is correlated with the amount of TGFβ2 present in AH, wherein non-OHT mice did not show increased active TGFβ2 levels, as seen in OHT mice (Figure 7D,H,K). Even though we used a congenic mice strain, it remains unclear why non-OHT mice did not produce sufficient active TGFβ2. Likewise, in some of the LV_CMV_eGFP-injected mice, the amount of eGFP fluorescence observed was small, spotted, and erratic around the TM (data not shown), indicating that the inconsistent TM transduction may result in insufficient TGFβ2 levels required for the desired pathological effect. It is possible that the accessibility of viral particles to the TM may vary from mouse-to-mouse. Recent studies in rats reported variability in the LV-induced GFP expression within their injected study cohorts [60]. Similar findings were observed in primary human TM cells that some strains produced enough active TGFβ2 in the conditioned medium and demonstrated cytoskeleton and ECM changes (Figure 8; Appendix A). This suggests that the effectiveness of LV_TGFβ2 to transduce TM cells of non-OHT mice may be lower compared to the OHT mice. 

In summary, we demonstrated the feasibility of a recombinant LV system to drive sufficient TGFβ2 expression with selective tropism to the TM and caused prolonged IOP elevation. The decrease in outflow facility was associated with OHT mice. These desired effects were achieved with no visible signs of inflammation or other ocular abnormalities in the anterior chambers. However, no loss in RGCs was observed in the timeframe studied. This limits the applicability of our model in studying glaucoma-associated optic neurodegeneration. Our study demonstrates that LV-mediated gene transfer is an effective tool for investigating TGFβ2-induced ocular hypertension in mice.

## 4. Methods and Materials

### 4.1. Mice

C57BL/6J (male) and BALB/cJ (female) mouse strains were obtained from Jackson Laboratories (Bar Harbor, ME, USA) and housed in a vivarium at the University of North Texas Health Science Center (UNTHSC, Fort Worth, TX, USA). All the following experimental procedures were executed in agreement with the guidelines and regulations of UNTHSC’s Institutional Animal Care and Use Committee (IACUC) and ARVO’s Statement for the Use of Animals in Ophthalmic and Vision Research. Mice between the age group of 15–25 weeks and weighing from 22–30 g were used for the study. Animals were fed standard chow ad libitum and kept in a 12 h light:12 h dark cycle under a controlled environment of 21–26 °C with 40–70% humidity. The number of mice used for every experiment is indicated in the respective figures.

### 4.2. Viral Vector Injections

We modified the hTGFβ2 sequence with the cysteine amino acid switched to serine at the 226 and 228 positions as described previously by Shepard et al. [41]. Lentiviral vector constructs encoding the active hTGFβ2 (C226, 228S) under the CMV promoter (LV_CMV_hTGFβ2^226/228^) were purchased from VectorBuilder (Product ID: VB170816-1094fnw; Chicago, IL, USA). Empty lentivirus vector (LV_CMV_Null) obtained from SignaGen Labs (Catalog #: SL100261; Frederick, MD, USA) was used as control, which was diluted at a 1:1 ratio with filtered 1× phosphate buffered saline (PBS) to obtain the desired viral load. Animals were examined by direct ophthalmoscopy (hand-held ophthalmoscope, model 11710; Welch-Allyn, Skaneateles Falls, NY, USA) to ensure that all eyes to be injected were not showing any detectable signs of ocular pathology. Before injections, the mice’s eyes were topically anesthetized by administering a drop of proparacaine HCl (0.5%) (Akorn Inc., Lake Forest, IL, USA). The ultra-pure lentiviral particles expressing TGFβ2 or null were intravitreally injected (2 × 10^6^ TU/eyes, 2 μL bolus) into contralateral eyes of mice anesthetized with isoflurane (2.5%; with 0.8 L/min oxygen) via the intranasal route. Intravitreal injections were executed using a Hamilton’s (Reno, NV, USA) glass micro-syringe attached with a 33 gauge 1-inch-long needle. The solution of virus particles was drawn into the syringe, ensuring no bubbles formed. The needle was positioned at an angle of ~40° from the limbus of the temporary proptosed mouse eye and carefully inserted into the vitreous chamber via the equatorial sclera at a depth of about 10–20 mm. Care was taken to avoid touching the needle to the posterior side of the lens and retina. The viral solution was slowly injected into each eye, and the needle was left inside the vitreous briefly for 1 min before being rapidly withdrawn from the eyes. 

### 4.3. IOP Measurements

For the data represented in Figure 3 and Figure 4, IOP was monitored every week during the daytime on conscious mice following injections and once prior to injection for baseline readings. The IOP measurements were noted using a TonoLab rebound tonometer (Colonial Medical Supply, Londonderry, NH, USA), as previously described [75]. IOPs were also monitored during daylight (between 9:00–11:00 am) and dark conditions (between 6:00–8:00 am) on mice under anesthetic influence via intranasal isoflurane (2.5%; 0.8 L/min oxygen) delivery [61]. The measurements were completed within 3 min of isoflurane influence to avoid any of its side effects on IOP. Six values per eye were recorded in a masked manner, and the average value was represented. 

### 4.4. Outflow Facility Measurements

Aqueous humor outflow facility (C) was assessed using the constant-flow infusion model formerly demonstrated by Millar et al. [76,77] in live mice. The xylazine/ketamine (10/100 mg/kg; Vetus; Butler Animal Health Supply, Westbury, NY/Fort Dodge Animal Health, Fort Dodge, IA, USA) cocktail was intraperitoneally administered to the mice, followed by one-quarter to one-half of the initial dose, as required, for continuous maintenance of the surgical anesthetic state. A heating pad at 37 °C was used throughout the procedure to preserve the physiological temperature of the mice under anesthesia. For corneal anesthesia, topical drops of proparacaine HCl (0.5%) (Akorn Inc.) were applied to each eye before cannulating the anterior chamber. The 30-gauge needle was inserted through the cornea 1–2 mm from the limbus, positioned parallel to the iris, and pushed towards the chamber angle opposite to the cannulation point. Care was taken to not touch the iris, corneal endothelium, or the anterior lens capsule. To prevent corneal drying, a drop of filtered saline was applied to each eye. Each cannulated needle was connected to a calibrated BLPR-2 flow-through pressure transducer (World Precision Instruments (WPI), Sarasota, FL, USA) via PE60 tubing to constantly monitor the pressure within the perfusion system. Another set of tubing connected the opposite terminal of the transducer to a 1 mL syringe loaded onto a micro-dialysis infusion pump (SP101I Syringe Pump; WPI), determining the flowrate. The entire perfusion system was carefully filled with sterile and filtered PBS, avoiding any air bubbles. The initial flowrate of 0.1 μL/min was used to infuse the eyes and stabilize the pressure, followed by a gradual increase in flowrate with a 0.1 μL/min increment up to 0.5 μL/min. The signals from the pressure transducer were amplified and converted to digital records displayed on the virtual charts of LabScribe2 software (WPI). Three stabilized pressures were recorded at 5 min intervals and averaged as the mean for each flowrate, then plotted as ordinate and abscissa, respectively. The reciprocal of the slope of this plot was computed as C for each eye. Mice were euthanized after this procedure. 

### 4.5. Ophthalmoscopy and Slit-Lamp Examination 

Eyes of all mice were clinically examined by direct ophthalmoscopy with a hand-held ophthalmoscope (model 11710; Welch-Allyn, Skaneateles Falls, NY, USA) to assess gross immunogenicity and lenticular opacity throughout the course of the experiment. Slit-lamp (SL-D7, Topcon Corporation, Tokyo, Japan) was used to determine inflammation and ocular abnormalities in the anterior segment, including corneal edema, and photo-documented with a digital camera (DC-4; Topcon) as described earlier [78].

### 4.6. Ocular Sample Collection 

Following intravitreal delivery of lentiviruses, animals were euthanized at specified timepoints with CO_2_ asphyxiation. For histological staining, each eye was immediately enucleated and placed in 4% paraformaldehyde (PFA, Electron Microscopy Sciences, Hatfield, PA, USA) for overnight fixation at 4 °C, followed by 1× PBS (Sigma-Aldrich Corp, St. Louis, MO, USA) washes, dehydration by ethanol, and embedded in paraffin wax for sectioning. For Western blot application, AH (~5 μL) was collected from the anterior chamber before enucleation. The eyes were firmly positioned in between forceps to access the anterior chamber, and any excess tear-film was gently dabbed from the corneal surface. The cornea was carefully punctured with a 32 gauge needle to collect the aqueous humor. The eyes were then enucleated to carefully bisect the anterior segment from the posterior section. Anterior segments inclusive of iridocorneal angle were immersed in the RIPA lysis buffer (150 μL; EMD Millipore, Billerica, MA, USA) and were thoroughly crushed using a conical glass-pestle to facilitate lysis of the tissue. The cell debris was separated from the lysate by centrifugation at 12,000× *g* rpm for 10 min. All the samples collected were placed on ice during the procedure or preserved at −80 °C till further use.

### 4.7. Lentiviral Transduction of Primary Human Trabecular Meshwork Cells 

Primary TM cells isolated from different donor human eyes were cultured as previously described (*n* = 4) [79]. The cells were maintained in DMEM media (Sigma-Aldrich) supplemented with 10% FBS (Gibco Life Technologies, Grand Island, NY, USA), 1% pen-strep (Gibco), and 1% glutamine (Gibco). The cells were trypsinized (TrypLE Express, Gibco) and plated equally in each well of 4-well chamber slides (Lab-Tek Nunc Brand Products, Rochester, NY, USA) at a confluency of 60–80%. Media was regularly changed every alternate day until about 95–100% confluency was achieved per well. Cells were then transduced (day 0) with either LV_CMV_hTGFβ2^C226/228S^ (VectorBuilder) or LV_CMV_Null (SignaGen Labs) vectors at a viral load of 5 MOI/mL/well with polybrene (7.5 μg/mL; VectorBuilder) in antibiotic-free media supplemented with 6% FBS. Post 2 days of transduction, the cells were switched back to a regular maintenance medium. Conditioned medium was collected after every 2 days and subjected to further analysis. Cells were fixed with 4% PFA (15 min) on day 11 followed by PBS washes for protein analysis via immunostaining.

### 4.8. H&E Staining 

Hematoxylin and eosin (H&E) staining was performed on 5 μm sagittal sections of paraffin-embedded mouse eyes to examine the general morphology of the anterior segment, including the TM structure at iridocorneal angle by light microscopy. Images were captured using a Keyence microscope (Itasca, IL, USA) at DIC (differential interference contrast) settings.

### 4.9. Immunostaining 

For paraffin-embedded mouse tissues, 5 μm sections were prepared, deparaffinized in xylene, and rehydrated with gradual 5 min washes in each 100, 95, 70, and 50% ethanol solution and ending with a 10 min wash in 1× PBS. The tissue sections selected for intracellular staining were subjected to antigen retrieval in citrate buffer (pH 6.0), heated in a 70 °C water-bath for 1 h, and cooled to room temperature for 30 min. The sections were blocked using 5% bovine serum albumin (BSA; Gene Clone, San Diego, CA, USA), 5% goat serum (EMD Millipore Corp), and 0.2% Triton X-100 (diluted in PBS; Fisher BioReagents, Fair Lawn, NJ, USA) for 2–3 h. The sections were briefly rinsed in 1× PBS, followed by adding TGFβ2 primary antibody (catalog #: ab36495; Abcam, Cambridge, MA, USA) in 50% blocking buffer (1:100 dilution) and incubated overnight at 4 °C in a dark-humidifying chamber. The sections were washed 4 times with 1× PBS before incubating with Alexa Fluor secondary antibody (1:500; Invitrogen, Life Technologies, Grand Island, NY, USA) at room temperature for 2 h. The slides were washed again and mounted with DAPI antifade mounting medium (Vectashield, Vector Laboratories Inc., Burlingame, CA, USA) [80]. Fixed primary human TM cells were blocked with 1× PBS buffer containing 10% goat serum and 0.1% Triton-X100 for 2 h at room temperature. The cells were then incubated overnight at 4 °C with a cocktail of the respective primary antibodies (as listed in Table 1) diluted in blocking buffer followed by appropriate Alexa Fluor secondary antibody dilution (1:500). Images were captured, processed, and quantified using a Leica SP8 confocal microscope and LAS-X software (Leica Microsystems Inc., Buffalo Grove, IL, USA). Tissue sections and TM cells stained without primary antibodies served as a negative control and were used to normalize the fluorescent intensities by background elimination. The anterior segment staining was quantified by drawing the region of interest around the TM area and represented as the unit of fluorescence intensity per μm^2^. For quantification of primary TM cell staining, six different non-overlapping areas of each treated well were imaged and quantified for fluorescence intensity. Of the 4 different human primary TM cells transduced with LV_TGFβ2, only two cells responded with significant TGFβ2 expression. These TGFβ2 responder cells were utilized further for quantitative analysis.

### 4.10. Western Blotting 

Equal amounts of protein obtained from the aqueous humor were processed with sample buffer (NuPAGE, Invitrogen, Life Technologies) to a final concentration of 1× at 100 °C for 6 min. After cooling down, the samples were loaded on a readymade denaturing gradient polyacrylamide gel (4–12%; NuPAGE Bis-Tris gels, Life Technologies, Grand Island, NY, USA) along with protein ladder (Invitrogen). The gels were immersed in 1× MOPS SDS running buffer (NuPAGE, Invitrogen) and run using Invitrogen’s Mini Gel electrophoresis tank at constant voltage (150 V). The separated protein gels were stacked together over a methanol-activated PVDF membrane (Immobilon-P, 0.45 μm pore size; Merk Millipore Ltd., St. Louis, MO, USA) in a blot module containing cold 1× transfer buffer (NuPAGE) with 20% methanol. The proteins were transferred onto the PVDF blots at constant voltage (24 V). Next, the membranes were blocked for 2 h at room temperature with 3% BSA in 1× Tris-buffered saline with Tween-20 (TBST) (for TGFβ2, Abcam) or 10% nonfat dry milk prepared in 1× phosphate buffered saline with Tween-20 (PBST) and were then incubated with respective primary antibodies (1:1000 dilution, Table 1) overnight at 4 °C on a rotator. The membranes were washed thrice with 1× TBST/PBST for 5 min each before incubating with respective horseradish-peroxidase (HRP)-conjugated secondary antibodies (1:2500) for 2 h at room temperature. The membranes were washed again (as described above) and developed with enhanced chemiluminescence (ECL) detection reagents (SuperSignal West Femto Maximum Sensitivity Substrate; Life Technologies) using an LI-COR Biosciences Odyssey-Fc image system (Lincoln, NE, USA), as previously explained [80,81]. The variability in AH may derive due to the mixture of impurities from the tear-film. The total proteins were measured as loading controls for AH samples either via Ponceau S stain (Sigma-Aldrich; before blocking the membrane) or Coomassie Brilliant Blue staining (R-250, Bio-Rad Labs, Hercules, CA, USA; after probing all the antibodies, and followed by stripping the blots). For conditioned media samples collected from the treated cells, 1 mL of the media was incubated with the 100 μL StrataClean Resin (Agilent Technologies, Santa Clara, CA, USA) overnight at 4 °C on constant rotation. The beads were then centrifuged and incubated with a 60 μL sample buffer for 6 min at 100 °C. The processed samples were centrifuged, and 40 μL of sample volume was carefully loaded onto the polyacrylamide gel to run the Western blot analysis as described above. The total protein load was determined by Coomassie staining.

### 4.11. Pattern Electroretinography 

Retinal ganglion cell (RGC) function was evaluated simultaneously on both eyes of the mice by measuring the amplitudes and latencies obtained from the pattern electroretinogram (PERG) readings as described previously [61,82]. The mice were briefly anesthetized with ketamine/xylazine (100/10 mg/kg) solution and placed on the regulated heating stage of the PERG apparatus for the entirety of the procedure, as described previously by Chou et al. [83]. Miami PERG system was used (Jorvec, Miami, FL, USA) to regulate the patterned visual stimuli and record PERG signals as per the manufacturer’s instructions. 

### 4.12. RGC Counts from Retinal Whole Mounts 

Retinal whole mounts were stained with RBPMS to examine and compute the surviving RGCs following the LV transduction, as described previously [61,78,84]. Eyes enucleated from the euthanized mice were fixed with 4% PFA for 15 h at 4 °C. The eyeballs were briefly rinsed with 1× PBS and carefully dissected to separate the anterior chamber and posterior cup from the retina. The isolated whole retinas were then incubated overnight with blocking buffer (10% goat serum with 0.2% Triton X-100) at 4 °C, followed by incubation in RBPMS antibody at 1:200 dilution (Gentex, Irvine, CA, USA; Catalog # GTX118619) for 2 days at 4 °C. Later, the retinas were washed for 3 h in PBS and incubated in appropriate secondary antibody (goat anti-rabbit-Alexa488, 1:500, Invitrogen) for 2 h at room temperature, and then washed again for 2 h in PBS. Four slits were made on the periphery of the retinal cups to facilitate their flat mounting on the slides with DAPI antifade mounting medium (VectaShield, Vector Laboratories Inc.). For RGC counting, at least 16 non-overlapping images of the entire retina (representing peripheral, mid-peripheral, and central area) were captured at 200× magnification using a Keyence fluorescence microscope (Itasca). RBPMS-positive cells were counted from these images using the ‘Macros’ function of ImageJ software [85]. The RGC size, shape, and fluorescent intensity (Alexa 488) were defined in the Macros system. Few images were randomly selected for manual counting to determine the accuracy of the Macros function.

### 4.13. Statistical Analysis 

Statistics for each experiment were calculated using Prism 9.0 software (GraphPad, San Diego, CA, USA) to determine the significance of the data that was less than the *p*-value of 0.05 for each analysis. Quantified data were presented as the mean ± SEM (standard error of the mean) with biological replicates of at least three (*n* ≥ 3) for both in vivo and in vitro studies. One-tailed paired Student’s *t*-test was used to compare results between two groups (control vs. treated). The IOP results that comprise more than two groups were analyzed by repeated-measures two-way ANOVA followed by a Bonferroni post-hoc test. Linear regression was evaluated by Pearson correlation using a one-tailed analysis for *p*-value.

## Figures and Tables

**Figure 1 ijms-23-06883-f001:**
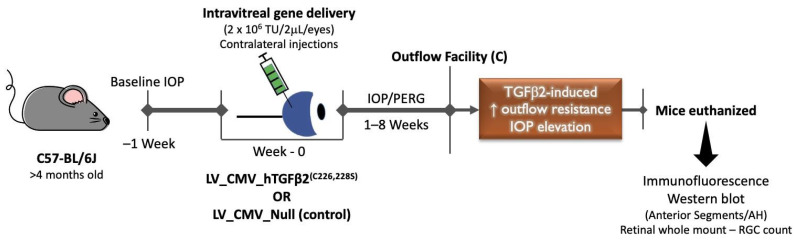
Schematic representation of experimental design for intravitreal injections of LV_TGFβ2. Baseline IOP was measured in 4 months old C57BL/6J mice, and a single intravitreal injection of LV_TGFβ2 in one eye and LV_Null in the contralateral eyes were performed. IOPs were monitored weekly, and the outflow facility was measured 8 weeks post-injection. After 7 to 8 weeks of injections, the PERG was measured, and mice were sacrificed to analyze RGC loss. Anterior segment or AH samples were isolated to further analyze TGFβ2 levels. LV—lentivirus; TGFβ2—transforming growth factor beta-2; CMV—cytomegalovirus promoter; IOP—intraocular pressure; PERG—pattern electroretinogram; AH—aqueous humor; RGC—retinal ganglion cells.

**Figure 2 ijms-23-06883-f002:**
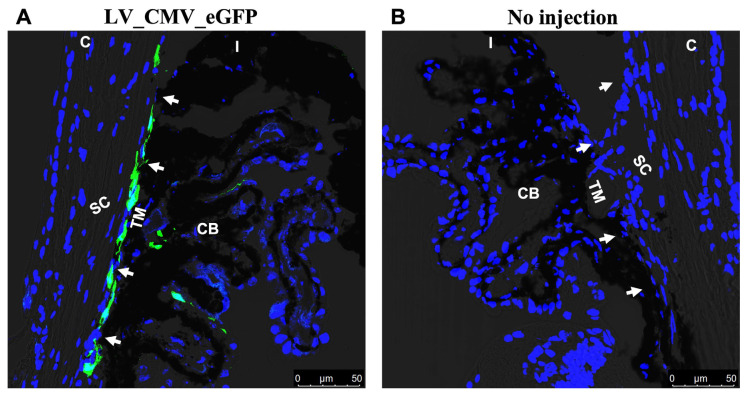
LV exhibits specific tropism for TM in mice. (**A**) C57BL/6J mice eye intravitreally injected with LV_CMV_eGFP (2 × 10^6^ TU/2 μL/eye bolus; *n* = 3) showed tropism towards the iridocorneal angle of the anterior chamber (representative image shown; scale 50 μm). The LVs induced substantial eGFP expression primarily in the TM region (represented by white arrows) post 3 weeks of injections. (**B**) Anterior segment section of contralateral un-injected eye showing no eGFP expression or background in the iridocorneal region. TM—trabecular meshwork; SC—Schlemm’s canal; CB—ciliary body; C—cornea; I—iris.

**Figure 3 ijms-23-06883-f003:**
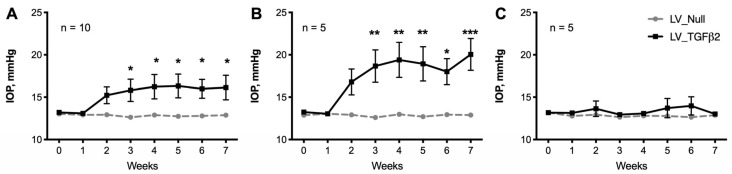
C57BL/6J mice developed ocular-hypertension when intravitreally injected with LV_TGFβ2 constructs. Intravitreal injections of LV_Null (control) and LV_TGFβ2 vectors (*n* = 10 each) were administered in contralateral eyes of C57BL/6J male mice following baseline IOP measurements. Weekly monitoring of daytime-conscious IOPs showed significant and sustained IOP elevation ((**A**), combined) 3 weeks post-injections. Mice with LV_TGFβ2-induced IOPs showing Δ change of >2.5 mmHg (**B**) were segregated from mice observed with no IOP elevation (**C**). Repeated measures two-way ANOVA with Bonferroni post-hoc analysis, data represented as mean ± SEM, * *p* < 0.05, ** *p* < 0.01, *** *p* < 0.001. (Δ—delta change).

**Figure 4 ijms-23-06883-f004:**
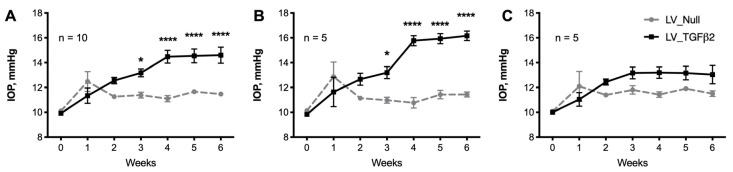
LV_TGFβ2 vectors caused IOP elevation in BALB/cJ mice. Conscious daytime IOPs were monitored weekly on BALB/cJ mice (*n* = 10) intravitreally injected with LV_Null and LV_TGFβ2 vectors in contralateral eyes. Baseline IOP was measured before injections (0 weeks) to ensure that each mouse had normal IOP before treatment. IOP was significantly elevated in LV_TGFβ2 eyes starting 3 weeks of injections ((**A**), combined). The (**B**) mice with Δ IOP change of >2.5 mmHg were separated from mice (**C**) with very low IOP elevation. Repeated measures two-way ANOVA with Bonferroni post-hoc analysis, data represented as mean ± SEM, * *p* < 0.05, ***** p* < 0.0001.

**Figure 5 ijms-23-06883-f005:**
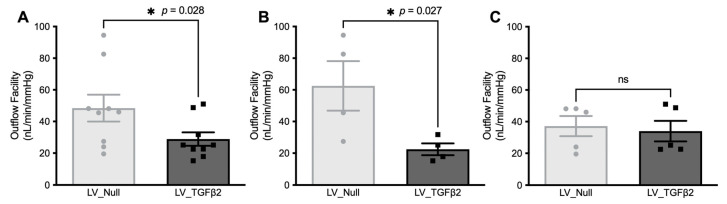
LV_TGFβ2 intravitreal injections decreased aqueous outflow facility in C57BL/6J mice. One eye of each mouse was injected with the TGFβ2^C226/228S^-expressing lentiviral vector (black squares), while the contralateral eye was injected with the Null-lentiviral vector (gray circles; *n* = 10; same cohort as Figure 3). AH outflow facility was evaluated on both eyes at 7 weeks after injection for mice with significant IOP elevation (*n* = 5) (**B**) and at 8 weeks after injection for mice with no IOP elevation (*n* = 5) (**C**). Mice were segregated based on individual assessment of elevated IOP measurements (refer to Figure 3). (**A**) The combined effect of all mice (*n* = 10). Paired (one-tailed) student *t*-test, data represented as mean ± SEM.

**Figure 6 ijms-23-06883-f006:**
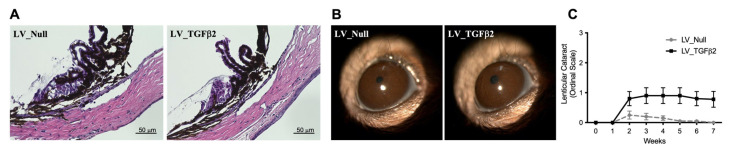
Effect of lentiviral infection on ocular structures of mice anterior segments. Contralateral eyes of C57BL/6J mice were intravitreally injected with LV_Null and LV_TGFβ2 vectors. (**A**) Representative hematoxylin and eosin (H&E) staining image of fixed and paraffin sectioned eyes, 9 weeks post-injections (*n* = 5; scale 50 μm). Average IOP measured for LV_Null was 14.12 mmHg (day) or 18.04 mmHg (dark; refer to Appendix A), and for LV_TGFβ2 was 18.23 mmHg (day) or 23.16 mmHg (dark). (**B**) Representative slit-lamp images of the anterior segments to determine immunogenicity and corneal edema. (**C**) Ophthalmoscopy scoring on an ordinal scale for lenticular cataract in C57BL/6J mice, *n* = 10 (same mice cohorts as described in Figure 3). No hyperemia was observed in these mice throughout the course of the experiment (data not shown). Repeated measures two-way ANOVA with Bonferroni post-hoc analysis, data represented as mean ± SEM.

**Figure 7 ijms-23-06883-f007:**
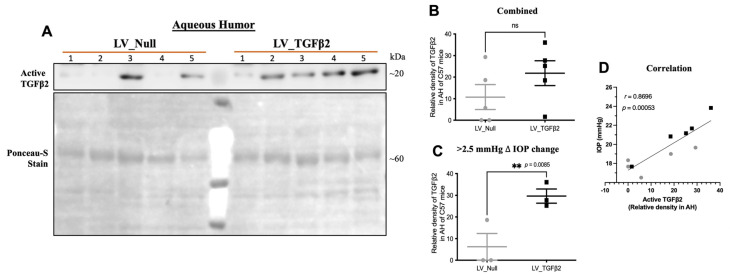
Intravitreally-delivered LV_TGFβ2-induced active TGFβ2 expression in anterior segments of injected eyes. Contralateral eyes of C57BL/6J mice were intravitreally injected with LV_Null (gray circles) and LV_TGFβ2 (black squares) vectors. (**A**) Western blot of AH samples collected 11 weeks post-injection from mouse anterior chamber and (**B**,**C**) their quantitative analysis normalized to Ponceau-S stain (*n* = 5). (**C**) Represents a significant increase in active TGFβ2 expression in eyes, showing > 2.5 mmHg of increase in IOP (*n* = 3). (**D**) Correlation of week 10 post-injection IOP with week 11 post-injection TGFβ2 relative density in AH (Pearson correlation). (**E**) Representative immunostaining image of fixed and paraffin sectioned eyes, 8 weeks post-injections (*n* = 5; scale 75 μm). The quantitative analysis of (**F**,**G**) TGFβ2 and (**I**,**J**) alpha smooth muscle actin (αSMA) intensity in the TM region only (represented by white arrows). (**G**) TGFβ2 and (**J**) αSMA expression in mice with significant IOP elevation (*n* = 3; refer to Appendix A for IOP data). (**H**,**K**) Correlation of week 8 post-injection IOP with week 9 post-injection (**H**) TGFβ2 or (**K**) αSMA fluorescent intensity in anterior tissue (Pearson correlation). (**D**,**H**,**K**) Black square data points represent LV_TGFβ2-injected eyes (*n* = 5), and gray circles represent LV_Null eyes (*n* = 5). Paired (one-tailed) student *t*-test, data represented as mean ± SEM. TM—trabecular meshwork; SC—Schlemm’s canal; CB—ciliary body; C—cornea; I—iris; R—retina.

**Figure 8 ijms-23-06883-f008:**
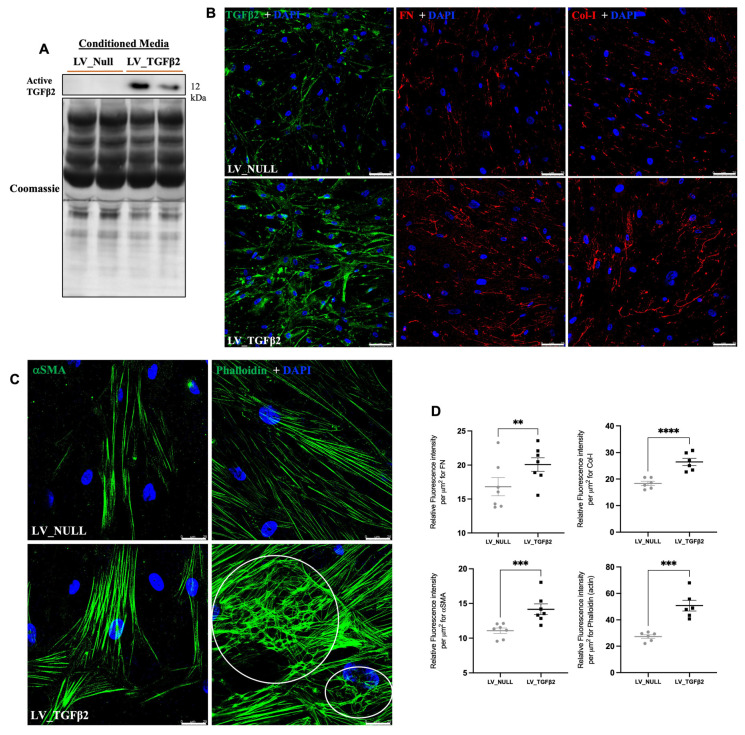
LV_TGFβ2 transduction induced extracellular matrix (ECM) and cytoskeletal alterations in primary human TM cells. Primary human TM cells were treated with 5 MOI viral load of LV_Null (gray circles) and LV_TGFβ2 (black squares) vectors and incubated for 11 days (*n* = 2 responder strain, representative images shown). (**A**) Active TGFβ2 levels were determined in conditioned media of the transduced cells. The total protein loading was determined via Coomassie staining of the PVDF membrane. (**B**) TGFβ2, ECM markers (fibronectin (FN) and collagen-I (Col-I)) (scale 75 μm), and (**C**) cytoskeletal changes (αSMA and F-actin) were also determined via immunostaining (scale 25 μm). LV_TGFβ2-induced expression of all the markers with morphological changes in the cells that included the formation of cross-linked actin networks (CLANs; marked within white circles) and F-actin stress fibers as depicted via phalloidin staining. (**D**) Quantitative analysis of FN Col-I, αSMA, and phalloidin staining from at least 6 different non-overlapping regions of the treated wells. Paired (one-tailed) student *t*-test, data represented as mean ± SEM, ** *p* < 0.01, **** p* < 0.001, ***** p* < 0.0001.

**Table 1 ijms-23-06883-t001:** Primary antibodies used for immunostaining or Western blot.

Antibodies	Catalog #	Company	Dilutions
Transforming Growth Factor-β2	Ab36495	Abcam, Cambridge, MA, USA	1:100
Fibronectin (FN)	Ab2413	Abcam	1:400
Collagen-1 (Col-1)	NB600-408	Novus Biologicals, Littleton, CO, USA	1:100
α-Smooth Muscle Actin (αSMA)	Ab7817	Abcam	1:100
Phalloidin-Alexa Conjugate	8878S	Cell Signaling Technology, Danvers, MA, USA	1:20

## Data Availability

All data that support the reported results are included in this manuscript and in the Appendix A.

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
