# Peer review of "A Novel Mouse Model of TGFβ2-Induced Ocular Hypertension Using Lentiviral Gene Delivery"

_ijms, 2022, doi:10.3390/ijms23136883_

Round 1

Reviewer 1 Report

The authors describe lentiviral-driven expression of TGFB2 in the trabecular meshwork of mice as a novel model of ocular hypertension. The study is generally well-executed but I have a few major recommendations to the authors that I think will improve the clarity of the manuscript. There are a couple of additional experiments and re-wording of grandiose statements that need attention.

  • Throughout the manuscript there are minor grammatical errors that need to be addressed. I will point out as many as I can but the authors should take care to address them more carefully throughout:
    1. Line 71 “an experimental model”
    2. Line 73 – mutations
    3. Line 255 – the authors leave (cite) in the manuscript – I assume a citation was needed here?
    4. Line 280 – grammatically incorrect “development of investigative model” – I have problems with this statement because the authors are not developing a model of glaucoma here – they do not adequately assess optic neuropathy here – only IOP and outflow.
    5. Line 374 – mouse to mouse, not mice- to mice
  • The introduction is a little lengthy and reads much more like a review paper. It needs to be shortened with authors taking care to convey concisely the most important points. I would reduce this section by about 1/3 for readability.
  • The introduction needs to be restructured – there is a lot of talk about glaucoma, which I understand may provide rationale for these experiments, but the authors do not adequately assess optic neuropathy and glaucoma pathophysiology (see below). Perhaps emphasis on trabecular meshwork function should be the story here.
  • For figure 2, can the authors increase the size of the scale bar, it is unclear.
  • I am surprised that after 9 weeks of elevated IOP that the authors see no effect on RGC function. Previous studies using microbeads to induce elevations, or episcleral vein cauterization have reported similar (~30% increase) in pressures and corresponding axon dropout (and RGC soma). I find it interesting that the authors do not see a drop-out of RGCs. This could be due to the methods used in the paper, for example, or the relatively low n. PERG analysis is notoriously variable and usually requires a larger n for more robust analysis. I am not sure how much information the authors can derive from the analysis carried out.
  • The authors don’t describe how RGCs were counted, only that they were “automated” in “ImageJ”. This is not enough detail for a manuscript, one cannot determine if the results were robustly or accurately measured with this information. I suggest the authors provide a more detailed description of the method used.
  • I do not feel that the authors have best characterized this model of ocular hypertension as it relates to optic neuropathy. A better way to determine if there was axonopathy associated with ocular hypertension would be to carry out axon counts in optic nerves. This should be done to assess damage in this model. This would better characterize the model.
  • Figure 6 – can the authors explain what the p-values refer to? They are also not mentioned in the text.
  • Figure 7 – it is not clear on first read of the manuscript what the difference is between the analysis carried out in 7B vs. C and E vs. F. Can the authors add a clear title to these plots? Can correlation analysis be done on each mouse showing IOP vs. TGF beta detected? This would be very interesting.
  • Section 2.5 – the authors carry out western blots of TM cell media to look at TGF beta levels. They then carry out IHC to look at levels of TGF beta and Collagen and Fibronectin. The IHC is not quantitative – is it representative of how many cells? How many wells? Some analysis needs to be done to support the conclusions by the authors. I suggest western blotting of cell lysates as the better choice. Otherwise, quantitative analysis of multiple cell images would improve this section.
  • Discussion – authors should elaborate more on lack of RGC pathophysiology – with additional nerve analysis and better RGC quantitation, this may change
  • Lines 330 – 332 – this is for EARLY glaucoma. The study that the authors have presented here does not assess early RGC function and axonopathy, this is not used in the right context here!! Yes, early RGC excitability acts to try to counter the IOP-induced pathology as a compensatory mechanism – it does not prevent degeneration. I suggest the authors re-read this paper and re-consider this statement.
  • Line 360 – where the authors discuss tissue stiffness due to ECM remodeling, can the authors make comments about how they could test the stiffness of the TM in future studies?
  • Final statement – “as an effective tool to model mouse strain for investigating……glaucomatous TM pathology” – is this really glaucoma??? You show no RGC effects???? Re-phrase.

Author Response

Dear Reviewers,

We appreciate the reviewer’s for providing valuable insights and feedback, which have substantially improved our manuscript. As per the suggestions, we have addressed all the concerns. Additional experiments and analyses were performed to address these concerns. These changes are highlighted either in the blue color or track changes in the revised manuscript. The major changes including new experiments are summarized below and described in detail later in response to specific concern raised by the reviewers.

  1. We have corrected our revised manuscript for grammatical and spelling errors.

  1. We have revised some of our statements to describe our model as ocular hypertension (OHT) model rather than glaucoma model as we have not reported any RGC loss. Introduction, discussion, and our concluding paragraph has been updated accordingly.

  1. Based on reviewer’s suggestion, we have elaborated discussion in revised manuscript to explain lack of RGC pathophysiology. We have also provided detailed description to explain methods and experimental setup used for analyzing RGC loss.

  1. We have now performed additional experiment which shows co-staining for TGFb2 and aSMA in the mouse anterior segments (Figure 7E-K). We have also included correlation analysis of IOP with TGFb2 expression in aqueous humor (Figure 7D) and anterior segments (Figure 7H) along with aSMA expression (Figure 7K).

  1. We have quantified immunostaining for ECM and cytoskeletal markers expressed in LV_TGFb2 transduced primary TM cells and presented as unit of fluorescent intensity per standard area in the revised manuscript (Figure 8D).

Please see the attachment for specific responses.

Reviewer 2 Report

The research manuscript is having good scientific potential. The author reported the sustained IOP elevation and low immunogenicity via lentiviral gene transfer of active TGFB2 transgene having TM region specificity. The manuscript needs to recheck for grammar and spelling. 

Author Response

(The authors gave the same response as above.)

Reviewer 3 Report

This is an well~written article of a new mouse model for Primary open angle glaucoma (POAG). Some points are listed below which would be revised for a better manuscript.

  1. Among the previous reported mouse model for POAG, what is a better characteristic point and what is a weak point?
  2. Is there any evidence that the induced TGF-b2 overexpression causes fibrosis around the area of injection as a characteristic of disease progression?
  3. Is there any degradation change or damage (via increased IOP) among the optical nerve in this model?
  4. If TGF-b2 is a significant inducer of POAG, it might be important to know the therapeutic timing to block TGF-b signaling. It might be important to use the inducible system of Lentiviral Gene Delivery system to identify the time points of no return (this is a suggestion).

Author Response

(The authors gave the same response as above.)

Reviewer 4 Report

It is well-designed study to induce OHT model in mice with less inflammatory reaction and more sustained IOP elevation period compared to previous adenoviral vector mediated TGF-b2-induced TM dysfunction model. The only limitation of this study is not to induce significant RGC loss and not to show any gliosis in the retina, it means that this model is not suitable for glaucoma model in mice but can be used for investigating TM degeneration induced by TGF-b related pathway.

Author Response

(The authors gave the same response as above.)
